# Calibrated ensembles can mitigate accuracy tradeoffs under distribution shift

**Ananya Kumar**[1]          **Tengyu Ma**[1]          **Percy Liang**[1]          **Aditi Raghunathan**[2]

[1]Computer Science Dept., Stanford University, Stanford, California, USA
[2]Computer Science Dept., Carnegie Mellon University, Pittsburgh, Pennsylvania, USA

## Abstract

We often see undesirable tradeoffs in robust machine learning where out-of-distribution (OOD) accuracy is at odds with in-distribution (ID) accuracy: a robust classifier obtained via specialized techniques such as removing spurious features often has better OOD but worse ID accuracy compared to a standard classifier trained via ERM. In this paper, we find that ID-calibrated ensembles—where we simply ensemble the standard and robust models after calibrating on only ID data—outperforms prior state-of-the-art (based on self-training) on both ID and OOD accuracy. On eleven natural distribution shift datasets, ID-calibrated ensembles obtain the best of both worlds: strong ID accuracy *and* OOD accuracy. We analyze this method in stylized settings, and identify two important conditions for ensembles to perform well both ID and OOD: (1) standard and robust models should be calibrated (on ID data, because OOD data is unavailable), (2) OOD has no anticorrelated spurious features.

## 1 INTRODUCTION

Machine learning models suffer large drops in accuracy in the presence of distribution shift where the test distribution is different from the training distribution. As ML systems are widely deployed, it is important to train models that achieve good accuracy on unforeseen, out-of-distribution (OOD) examples. For example, models trained on medical data from a few hospitals should work well when deployed broadly [Zech et al., 2018, AlBadawy et al., 2018]. Similarly, when predicting poverty from satellite imagery, models trained on data from a few countries should work well on all countries, particularly those where labels are scarce due to resource constraints [Jean et al., 2016]. There has been a

lot of research interest in tackling this robustness problem under various settings such as robustness to spurious correlations [Heinze-Deml and Meinshausen, 2017, Sagawa et al., 2020a], domain generalization [Arjovsky et al., 2019, Sun and Saenko, 2016], demographic shifts [Hashimoto et al., 2018, Duchi et al., 2019] among others.

Across many of these settings, an unfortunate tradeoff arises: robustness interventions, such as removing spurious features or lightweight fine-tuning, typically improve the OOD accuracy but cause a drop in the in-distribution (ID) accuracy on new test points from the original distribution. This tradeoff is a major hurdle in using the multitude of proposed methods that aim to improve OOD accuracy. In practice, most inputs are likely to be ID, so it is unsatisfactory to use a robust model that has high OOD accuracy but performs less accurately on these majority ID points. On the other hand, standard models (trained without robustness interventions) can fail in the presence of even small shifts, and it can be dangerous to use a standard model even if OOD points are rare. In this work, we ask: *is there a general strategy to harness the strengths of both the standard and robust model to achieve high accuracy both ID and OOD, without using OOD data?*

We find that ID-calibrated ensembles, a simple approach of first calibrating the standard and robust models on only ID data and then ensembling them, outperforms prior state-of-the-art both ID and OOD. As illustrated in Figure 1, across 11 natural distribution shift datasets (e.g. geographical shift, style shift, subpopulation shift), ID-calibrated ensembles get the *best of both worlds*: strong ID accuracy of the standard model and robust accuracy of the OOD models. Averaged across these datasets, ID-calibrated ensembles achieve an ID accuracy of 90.3% (vs. 88.7% for the standard model and 86.8% for the robust model) and OOD accuracy of 74.5% (vs. 65.2% for the standard model and 72.3% for the robust model).

We then analyze when and why ID-calibrated ensembles can get the best of the standard and robust models, under a

*Accepted for the 38th Conference on Uncertainty in Artificial Intelligence* (UAI 2022).

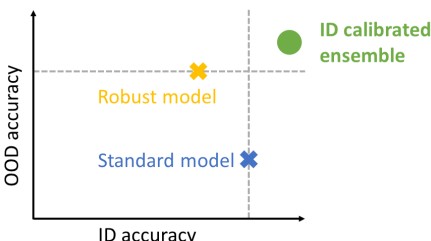

| | ID accuracy | OOD accuracy |
|---|---|---|
| Standard Model | 88.7% | 65.2% |
| Robust Model | 86.8% | 72.3% |
| ID calibrated ensemble (Std. + Rob.) | **90.3%** | **74.5%** |

(a) Plot of performance of different methods      (b) Performance of different methods averaged over 11 natural distribution shifts

Figure 1: In many settings, we have a standard model that performs better in-distribution, and a robust model that performs better out-of-distribution. Across 11 natural distribution shifts, ID-calibrated ensembles get the best of both worlds: the strong ID accuracy of the standard model and OOD accuracy of the robust model. We analyze its strengths and limitations in Section 4—as predicted by our analysis, ID calibrated ensembles do not perform as well on adversarially synthesized shifts with "anticorrelated" spurious features. We show full experimental results and ablations in Section 6.

simplifying assumption that these models provide different and independent signals for the label. If the standard and robust models are *calibrated* ID, the ensembling strategy for the best ID performance is to simply add the predictions of the two models (Proposition 4.1). By the same idea, if the standard and robust models were also calibrated OOD, ensembling would achieve the optimal OOD accuracy. However, since we only have ID training data, models can only be calibrated ID and ID calibration is not sufficient for OOD calibration [Ovadia et al., 2019].

When can calibrated ensembles provide benefits even without OOD calibration? In many natural distribution shifts, standard models pick up on predictive signals in the training data that are absent or suppressed under distribution shift—in these cases, we show that ID-calibrated ensembles obtain the best of both the standard and robust models OOD. However, when spurious features become anticorrelated OOD (as is common when the distribution shift is adversarially synthesized), we show that the ensemble's OOD accuracy is in between the standard and robust models. We empirically validate this on three adversarially synthesized shifts [Sagawa et al., 2020a, Jones et al., 2021] where the spurious signals are anticorrelated OOD.

Finally, we compare ID-calibrated ensembles to a number of other alternate ensembling strategies (for example, tuning the weights of the ensemble on ID validation data) and find that they do not work as well as ID-calibrated ensembles.

To summarize, our main contributions are:

1. We revisit the classic idea of ensembling and propose a simple, general, and effective method (ID-calibrated ensembles) to mitigate ID-OOD accuracy tradeoffs. This method outperforms prior approaches based on self-training, despite not using any additional unlabeled data.

2. We prove that ensembles of calibrated models are optimal when the models provide independent signals about the label. However, models can only be cali-

brated ID from which we have training data, and ID calibration does not imply OOD calibration. In simple and stylized settings, we identify conditions under which ID-calibrated ensembles achieve the best of standard and robust models in terms of OOD performance. We validate these insights experimentally and find that ID-calibrated ensembles eliminate tradeoffs under a variety of natural distribution shifts, but can fail when there are adversarially synthesized shifts.

## 2 SETUP

Consider a $K$-class classification task, where the goal is to predict labels $y \in [K]$ corresponding to inputs $x \in \mathcal{X}$.

**Models.** A model $f : \mathcal{X} \to \mathbb{R}^K$ takes an input $x \in \mathcal{X}$ and outputs a score $f(x) \in \mathbb{R}^K$ where $f(x)_i$ can be interpreted as the model's "confidence" that the label $y$ is $i$. The model outputs the label $\text{pred}(f(x)) = \arg\max_i f(x)_i$. The confidence scores can be normalized to sum to 1 (and interpreted as probabilities) using the softmax function, $\text{softmax}(f(x))_i = \frac{\exp(f(x)_i)}{\sum_{j=1}^{K} \exp(f(x)_j)}$ for $i \in [K]$.

**Distributions and error.** Let $P_{\text{id}}$ and $P_{\text{ood}}$ denote the underlying distribution of $(x, y)$ pairs in-distribution (ID) and out-of-distribution (OOD), respectively. We evaluate a model $f$ on the fraction of times it makes a wrong prediction on $P_{\text{id}}$ and $P_{\text{ood}}$: $\text{Err}_{\text{id}}(f) = \mathbb{E}_{x,y \sim P_{\text{id}}}[\text{pred}(f(x)) \neq y]$ and $\text{Err}_{\text{ood}}(f) = \mathbb{E}_{x,y \sim P_{\text{ood}}}[\text{pred}(f(x)) \neq y]$.

**Standard and robust models.** A standard model $f_{\text{std}}$ is trained via empirical risk minimization where we minimize some loss on ID training data. $f_{\text{std}}$ often relies on spurious correlations such as image background or occurence of certain words that are not necessarily predictive OOD. In order to improve OOD performance, a robust model $f_{\text{rob}}$ is trained via a modified training procedure (robustness interventions) to discourage models from relying on ID-specific spurious features. Formally, we have the following

---

**Algorithm 1** ID-calibrated ensembles

---

**Require:** in-distribution validation data $\{(x_i^{\mathsf{val}}, y_i^{\mathsf{val}})\}_{i=1}^{n_{\mathsf{val}}} \sim P_{\mathsf{id}}$,
  standard and robust models $f_{\mathsf{std}}, f_{\mathsf{rob}} : \mathcal{X} \to \mathbb{R}^K$
  1: $\overline{f}_{\mathsf{std}}$ = Calibrate $f_{\mathsf{std}}$ on in-distribution (ID) data
  2: $\overline{f}_{\mathsf{rob}}$ = Calibrate $f_{\mathsf{rob}}$ on in-distribution (ID) data
  3: Return $f_{\mathsf{ens}}(x) = \frac{1}{2}(\text{softmax}(\overline{f}_{\mathsf{std}}(x)) + \text{softmax}(\overline{f}_{\mathsf{rob}}(x)))$

---

relationship between $f_{\mathsf{std}}$ and $f_{\mathsf{rob}}$.

$$\text{Err}_{\mathsf{id}}(f_{\mathsf{std}}) \leq \text{Err}_{\mathsf{id}}(f_{\mathsf{rob}}); \quad \text{Err}_{\mathsf{ood}}(f_{\mathsf{rob}}) \leq \text{Err}_{\mathsf{ood}}(f_{\mathsf{std}}). \tag{2.1}$$

The precise robustness intervention depends on the task—in Section 4 we model the relationship between $f_{\mathsf{std}}$ and $f_{\mathsf{rob}}$ in a stylized setting amenable for analysis, and in Section 5 we describe what $f_{\mathsf{std}}$ and $f_{\mathsf{rob}}$ are in our real datasets.

**Best of both worlds.** Our goal is to get the best of both worlds—a classifier $f_{\mathsf{ens}}$ that achieves the strong ID accuracy of the standard model, and OOD accuracy of the robust model:

$$\text{Err}_{\mathsf{id}}(f_{\mathsf{ens}}) \leq \text{Err}_{\mathsf{id}}(f_{\mathsf{std}}); \quad \text{Err}_{\mathsf{ood}}(f_{\mathsf{ens}}) \leq \text{Err}_{\mathsf{ood}}(f_{\mathsf{rob}}). \tag{2.2}$$

**ID validation data.** To get the best of both worlds, we only allow access to ID validation data, $\{(x_i^{\mathsf{val}}, y_i^{\mathsf{val}})\}_{i=1}^{n_{\mathsf{val}}} \sim P_{\mathsf{id}}$, for tuning hyperparameters. Following Xie et al. [2021], Koh et al. [2021], Gulrajani and Lopez-Paz [2020] we do *not* use any OOD validation data.

## 3 METHODS

**Proposed method: ID-calibrated ensembles.** Given a standard model $f_{\mathsf{std}}$ and robust model $f_{\mathsf{rob}}$, we first calibrate each model on the *in-distribution* validation data, and then add up their predictions (Algorithm 1). In our experiments, we calibrate using temperature scaling [Guo et al., 2017] with the cross-entropy loss $\ell$:

$$T_{\mathsf{std}} = \arg\min_T \frac{1}{n_{\mathsf{val}}} \sum_{i=1}^{n_{\mathsf{val}}} \ell\Big(\frac{\text{softmax}(f_{\mathsf{std}}(x_i^{\mathsf{val}}))}{T}, y_i^{\mathsf{val}}\Big) \tag{3.1}$$

$$T_{\mathsf{rob}} = \arg\min_T \frac{1}{n_{\mathsf{val}}} \sum_{i=1}^{n_{\mathsf{val}}} \ell\Big(\frac{\text{softmax}(f_{\mathsf{rob}}(x_i^{\mathsf{val}}))}{T}, y_i^{\mathsf{val}}\Big) \tag{3.2}$$

We then ensemble the two models by adding up the probabilities that they predict [Lakshminarayanan et al., 2017].

$$f_{\mathsf{ens}}(x) = \frac{1}{2}\Big(\text{softmax}\Big(\frac{f_{\mathsf{std}}(x)}{T_{\mathsf{std}}}\Big) + \text{softmax}\Big(\frac{f_{\mathsf{rob}}(x)}{T_{\mathsf{rob}}}\Big)\Big), \tag{3.3}$$

where the predicted label is $\text{pred}(f_{\mathsf{ens}}(x)) = \arg\max_y f_{\mathsf{ens}}(x)_y$.

**Ablations.** In Section 6 we ablate each component of the method, for example the calibration step, way of combining the models, and we compare to (calibrated) ensembles of two standard models, or of two robust models.

## 4 INTUITIONS AND ANALYSIS

In this section, we build basic intuitions for when and why ID-calibrated ensembles can get the best of both worlds (good ID accuracy of $f_{\mathsf{std}}$ and OOD accuracy of $f_{\mathsf{rob}}$), even without using any OOD data. We first define a stylized setting, and then analyze the ID performance in Section 4.1 and OOD performance in Section 4.2.

**Diverse features.** An intuitive and illustrative conceptual setting is the following: we assume inputs have some robust features (that are predictive both ID and OOD) and some spurious features (that are only predictive ID). $f_{\mathsf{std}}$ relies on the spurious features while $f_{\mathsf{rob}}$ relies on the robust features, both of which provide independent signals on the label.

**Assumption 4.1.** *We assume that $f_{\mathsf{rob}}$ and $f_{\mathsf{std}}$ have diverse features with respect to $P_{\mathsf{id}}$ and $P_{\mathsf{ood}}$, that is,*

$$f_{\mathsf{rob}}(x) \perp f_{\mathsf{std}}(x) \mid y \quad \text{when } (x, y) \sim P \text{ for } P \in \{P_{\mathsf{id}}, P_{\mathsf{ood}}\} \tag{4.1}$$

**Connection with prior assumptions.** The diverse features assumption is weaker than the assumptions in prior conceptual models of distribution shifts Chen et al. [2020], Sagawa et al. [2020b], Nagarajan et al. [2020] where robust and spurious features are disjoint parts of the input, each generated independently based on the label. In our setting, the features can be complicated functions of the inputs.

**Ensemble.** The ensemble $f_{\mathsf{ens}}$ simply adds up the predictions of the standard model $f_{\mathsf{std}}$ and robust model $f_{\mathsf{rob}}$. This is slightly different from Section 3, but is more amenable to analysis.

$$f_{\mathsf{ens}}(x) = f_{\mathsf{std}}(x) + f_{\mathsf{rob}}(x) \tag{4.2}$$

**Class-balanced.** For simplicity of exposition, we assume the class-balanced setting where every label $P(Y = y)$ is equally likely. Formally, we say $P$ is class-balanced if $P(Y = y) = 1/K$ for all $y \in [K]$. We analyze the general setting in Appendix A.

### 4.1 ID PERFORMANCE OF ENSEMBLES

In this section, we show that if $f_{\mathsf{std}}$ and $f_{\mathsf{rob}}$ are *calibrated* with respect to $P_{\mathsf{id}}$, then the ensemble $f_{\mathsf{ens}}$ is the best way to combine their predictions. Since we have access to validation data from $P_{\mathsf{id}}$, the first step of our method (Section 3) is to calibrate $f_{\mathsf{std}}$ and $f_{\mathsf{rob}}$ ID. We conclude the section by giving intuition for why this calibration step can be particularly important for deep neural networks.

Intuitively, calibration means that the probability that a model outputs for an event reflects the true frequency of that event: if a model says 1,000 patients have the flu with probability 0.1, approximately 100 of them should indeed have the flu. Formally, we look at joint calibration [Murphy, 1973, Brocker, 2009] where a model $f$ is calibrated with respect to a distribution $P$ if for all $x \in \mathcal{X}, y \in [K]$:

$$P(y \mid f(x)) = \text{softmax}(f(x))_y \qquad (4.3)$$

The following proposition says that if $f_{\text{std}}$ and $f_{\text{rob}}$ are calibrated on $P_{\text{id}}$, then $f_{\text{ens}}$ has lower error on $P_{\text{id}}$ than any other way of combining the two models—this also implies that $f_{\text{ens}}$ gets higher accuracy than $f_{\text{std}}$ and $f_{\text{rob}}$.

**Proposition 4.1.** *Suppose that $f_{\text{std}}$ and $f_{\text{rob}}$ are calibrated with respect to $P_{\text{id}}$, and that $P_{\text{id}}$ is class-balanced. Let $h : \mathbb{R}^K \times \mathbb{R}^K \to \mathbb{R}^K$ be an arbitrary function that combines the standard and robust model's predictions, and let $f_h$ be the resulting classifier: $f_h(x) = h(f_{\text{std}}(x), f_{\text{rob}}(x))$. The ensemble is better than any such combination classifier $f_h$: $Err_{\text{id}}(f_{\text{ens}}) \leq Err_{\text{id}}(f_h)$.*

The proof of Proposition 4.1 is in Appendix A. Intuitively, since $f_{\text{rob}}(x) \perp f_{\text{std}}(x) \mid y$, the Bayes optimal predictor is proportional to multiplying their predicted probabilities, which is equal to adding logits (logits are in log space). Proposition 4.1 has an important condition: the two models must be calibrated. In practice, deep learning models are miscalibrated [Guo et al., 2017], so our first step (Section 3) is to calibrate the models ID. We explain why the ID calibration step is important for deep neural networks.

**Why neural networks are miscalibrated.** Deep neural networks are typically large enough to memorize the training dataset, and are encouraged to magnify their weights (and hence their confidence) to decrease the training loss [Mukhoti et al., 2020, Bai et al., 2021]. The extent of this miscalibration and overconfidence depends on the training procedure [Hendrycks et al., 2019, Desai and Durrett, 2020]. In our case $f_{\text{std}}$ and $f_{\text{rob}}$ are trained in different ways and have different calibration (Appendix B.3).

**Why this miscalibration can hurt ensembling.** Concretely, consider two models $f'_{\text{std}}$ and $f'_{\text{rob}}$ which are calibrated on $P_{\text{id}}$. Let $f_{\text{std}}(x) = M f'_{\text{std}}(x)$ for large $M \in \mathbb{R}$ (this magnifies its weights as discussed above), and let $f_{\text{rob}} = f'_{\text{rob}}$. $f_{\text{std}}$ and $f'_{\text{std}}$ have the same predictions and therefore accuracy but $f_{\text{std}}$ is highly miscalibrated. The ensemble is then given by $f_{\text{ens}}(x) = M f'_{\text{std}}(x) + f'_{\text{rob}}(x)$. For very large $M$, $f_{\text{ens}}$ and $f'_{\text{std}}$ have the same predictions—this means that $\text{Err}_{\text{ood}}(f_{\text{ens}}) = \text{Err}_{\text{ood}}(f_{\text{std}}) < \text{Err}_{\text{ood}}(f_{\text{rob}})$, and so ensembling does not get the best of both worlds.

## 4.2 OOD PERFORMANCE OF ENSEMBLES

We showed that if $f_{\text{std}}$ and $f_{\text{rob}}$ are calibrated on a distribution $P$, then $f_{\text{ens}}$ is better than both models on $P$. However, our validation data is from $P_{\text{id}}$, so we can only calibrate $f_{\text{std}}$ and $f_{\text{rob}}$ ID. Even after this ID calibration step, $f_{\text{std}}$ and $f_{\text{rob}}$ are very miscalibrated OOD (on $P_{\text{ood}}$—see Appendix B.3 and Ovadia et al. [2019]).

Our goal in this section is to build basic intuitions for when ID-calibrated ensembles can get high OOD accuracy. We draw inspiration from distribution shift benchmarks but examine simplified and stylized shifts. A toy version of our analysis is visualized in Figure 2, where the standard model relies on spurious features that change out-of-distribution. If these features are "suppressed" or "missing" OOD, then $f_{\text{ens}}$ does better than $f_{\text{std}}$ and $f_{\text{rob}}$ (Figure 2b). However, if these features are anticorrelated OOD (correlated with the opposite label) then the accuracy of $f_{\text{ens}}$ is between $f_{\text{std}}$ and $f_{\text{rob}}$ (Figure 2c). We begin by formalizing these shifts, and then analyze the accuracy under these shifts.

**Missing spurious.** For our first setting, we draw inspiration from some distribution shift benchmarks. Consider Breeds Living-17 [Santurkar et al., 2020] where the goal is to classify an image as one of 17 animal categories. The category 'bear' in the ID training data contains images of black bears and sloth bears while the OOD dataset has images of brown bears and polar bears. A standard model trained on the ID dataset might latch onto very specific features about sloth bears (for example the presence of a shaggy mane) which are simply missing in the OOD dataset ($f_{\text{std}}(x) = 0$). A robust model could be trained to project these features out [Xie et al., 2021], so its predictions are still fairly reliable OOD.

**Definition 4.1** (missing spurious). *A distribution $P_0$ has missing spurious features if for $x \sim P_0$, we have $f_{\text{std}}(x) = 0$ almost surely and for some $\alpha \in \mathbb{R}^+$, $P_0(Y = y \mid f_{\text{rob}}(X) = f_{\text{rob}}(x)) = \text{softmax}(\alpha f_{\text{rob}}(x))_y$ for all $x \in \mathcal{X}$.*

**Suppressed features.** In some datasets, such as satellite remote sensing datasets [Jean et al., 2016, Xie et al., 2021], a standard model can latch onto country-specific features that may be less prevalent OOD.

**Definition 4.2** (suppressed features). *A distribution $P_\tau$ is said to have suppressed features if $P_\tau(Y = y \mid f(X) = f(x)) = \text{softmax}(\tau f(x))_y$ for all $x \in \mathcal{X}$ and $f \in \{f_{\text{std}}, f_{\text{rob}}\}$, where $\tau \in \mathbb{R}^+$.*

**Anticorrelated spurious.** In some settings, the spurious feature can be correlated with a label ID but *anticorrelated* OOD. For example, in Waterbirds [Sagawa et al., 2020a], the task is to classify if an image contains a waterbird or a landbird where in the ID dataset, waterbirds are primarily featured with water backgrounds and landbirds with land backgrounds, but in the OOD datasets the backgrounds are

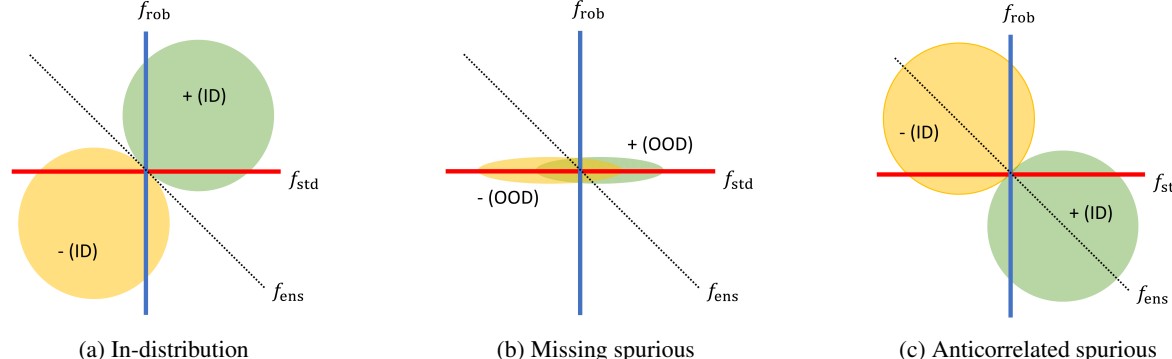

|  (a) In-distribution | (b) Missing spurious | (c) Anticorrelated spurious |

Figure 2: A toy version of our analysis in Section 4. (Figure 2a) Given a standard model $f_{\text{std}}$ (red horizontal line) and robust model $f_{\text{rob}}$ (blue vertical line) that use different aspects of the data, ensembling their predictions gives a predictor $f_{\text{ens}}$ (black dotted line) with lower error—in this case $f_{\text{ens}}$ completely separates the positive (green circle) and negative (yellow circle) examples in-distribution (ID). (Figure 2b) $f_{\text{std}}$ uses spurious features, suppose that these features are missing OOD (e.g., the $y$ component of the input goes close to 0)—then $f_{\text{std}}$ fares poorly and mislabels half the inputs, but the ensemble $f_{\text{ens}}$ is about as accurate as the robust model $f_{\text{rob}}$. (Figure 2c) On the other hand, suppose the spurious features are *anticorrelated* with the label OOD. In this case $f_{\text{ens}}$ intersects the positive (yellow circle) and negative (green circle) distributions, and gets 50% error—here $f_{\text{ens}}$ is worse than $f_{\text{rob}}$ but better than $f_{\text{std}}$.

flipped such that landbirds occur with water backgrounds and vice versa. This motivates the final definition of spurious shifts where the spurious features (background) are anticorrelated with the label OOD.

**Definition 4.3** (anticorrelated spurious)**.** *A distribution $P_{\text{adv}}$ is said to be anticorrelated spurious if for some $\alpha, \beta > 0$, for all $x \in \mathcal{X}$, $P_{\text{adv}}(Y = y | f_{\text{std}}(x)) = softmax(-\beta f_{\text{std}}(x))_y$ (note the minus sign), while $P_{\text{adv}}(Y = y \mid f_{\text{rob}}(x)) = softmax(\alpha f_{\text{rob}}(x))_y$.*

If the OOD distribution is a mixture of suppressed features and missing spurious features, then the ensemble $f_{\text{ens}}$ gets the best of both worlds.

**Proposition 4.2.** *If the OOD contains a mixture of suppressed features and missing spurious features i.e., $P_{\text{ood}} = \alpha P_\tau + (1 - \alpha)P_0$, and $P_\tau$ and $P_0$ are class-balanced, then we have $Err_{\text{ood}}(f_{\text{ens}}) \leq Err_{\text{ood}}(f_{\text{rob}})$ and $Err_{\text{ood}}(f_{\text{ens}}) \leq Err_{\text{ood}}(f_{\text{std}})$.*

On the other hand, if the OOD distribution contains anticorrelated spurious features, then the accuracy of $f_{\text{ens}}$ is in between the standard and robust models.

**Proposition 4.3.** *If spurious features are anticorrelated OOD so that $P_{\text{ood}} = P_{\text{adv}}$, then even if $P_{\text{adv}}$ is class-balanced, $Err_{\text{ood}}(f_{\text{rob}}) \leq Err_{\text{ood}}(f_{\text{ens}}) \leq Err_{\text{ood}}(f_{\text{std}})$.*

The full proofs appear in Appendix A.

## 5   DATASETS

We consider thirteen standard datasets, spanning multiple robustness interventions, types of shifts, and modalities (vision, language, time-series). We first describe the robustness interventions we consider, and then describe the datasets and types of shifts. All the datasets have been used by prior works on robustness, so we use their model checkpoints for reliable comparisons. See Appendix B.1 for more details.

**Robustness interventions**:

1. In-N-Out: Xie et al. [2021] use domain knowledge to project out spurious features in the input, and do an additional pretraining step. They call this robust model "aux-out" and show that it improves accuracy OOD, but hurts accuracy ID, compared to ERM.

2. Lightweight fine-tuning: We take checkpoints from Kumar et al. [2022] where the standard model fine-tunes all parameters on an ID dataset, and the robust model only learns the top linear 'head' layer (which does better OOD but worse ID).

3. Zero-shot language prompting: CLIP [Radford et al., 2021] is a multi-modal model that can predict the label of an image by comparing the image embedding, with prompts such as 'photo of an apple'. They show that this zero-shot language prompting approach (robust model) is more accurate OOD than fine-tuning the entire model (standard model), although ID accuracy of the robust model is worse.

4. Group distributionally robust optimization (DRO) [Sagawa et al., 2020a]: Standard ERM models often latch on to spurious correlations in a dataset, such as image background color, or the occurrence of certain words in a sentence. Group DRO

essentially upweights examples where this spurious correlation is not present.

5. CORAL [Sun and Saenko, 2016] aims to align feature representations across different domains, by penalizing differences in the means and covariances of the feature distributions. The hope is that this generalizes better to OOD domains.

We consider three types of natural shifts (geography shifts, subpopulation shifts, style shifts), and we also consider adversarially synthesized "anticorrelated" spurious shifts.

**Geography shifts.** In geography shifts the ID data comes from some locations, and the OOD data comes from a different set of locations. One motivation is that in many developing areas training data may be unavailable because of monetary constraints [Jean et al., 2016].

1. **LandCover** [Rußwurm et al., 2020]: The goal is to classify a satellite image into one of 6 land types (e.g., "grassland", "savannas"). The ID data contains images from outside Africa, and the OOD data consists of images from Africa. Xie et al. [2021] use the In-N-Out intervention.

2. **Cropland** [Wang et al., 2020]: The goal is to predict whether a satellite image is of a cropland or not. The ID dataset contains images from Iowa, Missouri, and Illinois, and the OOD dataset contains images from Indiana and Kentucky. Xie et al. [2021] use the In-N-Out intervention.

3. **iWildCam** [Beery et al., 2020, Koh et al., 2021]: The goal is to classify the species of an animal given a photo taken by a camera placed in the wild. The ID dataset consists of photos taken by over 200 cameras, and the OOD dataset consists of photos taken by held-out cameras placed in different locations. Koh et al. [2021] use the CORAL intervention.

**Subpopulation shifts.** In subpopulation shifts, the ID data contains a few sub-categories (e.g., black bear and sloth bear), and the OOD data contains different sub-categories (e.g., brown bears and polar bears) of the same parent category (e.g., bears). For both datasets below, Kumar et al. [2022] use the lightweight fine-tuning intervention.

1. **Living-17** [Santurkar et al., 2020]: the goal is to classify an image as one of 17 animal categories such as "bear", where the ID and OOD datasets have different species of bears.

2. **Entiy-30** [Santurkar et al., 2020]: similar to Living-17, except the goal is to classify an image as one of 30 entity categories such as "food", "motor vehicle", and "insect".

**Style shifts.** In style shifts, the ID data has a certain style (e.g., sketches), and the OOD data has a different style (e.g., real photos, renditions).

1. **DomainNet** [Peng et al., 2019]: a standard domain adaptation dataset. Here, our ID dataset contains "sketch" images (e.g., drawings of apples, elephants, etc), and the OOD dataset contains "real" photos of the same categories. Kumar et al. [2022] use the lightweight fine-tuning intervention.

2. **CelebA** [Liu et al., 2015]: the goal is to classify a portrait of a face as "male" or "female" - the ID dataset contains images of people without hats, and the OOD dataset contains images of people wearing hats (some facial features might be "suppressed" or "missing" with hats). Xie et al. [2021] use the In-N-Out intervention.

3. **CIFAR->STL**: standard domain adaptation dataset [French et al., 2018], where the ID is CIFAR-10 [Krizhevsky, 2009], and the OOD is STL [Coates et al., 2011]. The task is to classify an image into one of 10 categories such as "dog", "cat", or "airplane". Kumar et al. [2022] use the lightweight fine-tuning intervention.

4. **ImageNet** [Russakovsky et al., 2015]: a large scale dataset where the goal is to classify an image into one of 1000 categories. Radford et al. [2021] use the zero-shot language prompting intervention. We evaluate on 3 standard OOD datasets: **ImageNetV2** [Recht et al., 2019],**ImageNet-R** [Hendrycks et al., 2020], and **ImageNet-Sketch** [Wang et al., 2019].

**Anticorrelated spurious shifts.** In these adversarially synthesized shifts, the ID dataset contains a feature that is correlated with a label, but this correlation is flipped OOD. Jones et al. [2021] use the group DRO intervention.

1. **Waterbirds** [Sagawa et al., 2020a]: The goal is to classify an image as a "waterbird" or "landbird". The dataset is synthetically constructed to have anticorrelated spurious features: "water" backgrounds are correlated with "waterbird" labels in the ID, but anticorrelated OOD.

2. **MNLI** [Williams et al., 2018]: The goal is to predict whether a hypothesis is entailed, contradicted by, or neutral to an associated premise. Sagawa et al. [2020a] partition the dataset so that "negation" words are correlated with the contradiction label ID but these words are anticorrelated with the contradiction label OOD.

3. **CivilComments** [Borkan et al., 2019]: The goal is to predict whether a comment is toxic or not. Jones et al. [2021] partition the dataset so that in the ID split mentions of a Christian identity are correlated with non-toxic comments, but in the OOD split mentions of a Christian identity are correlated with a toxic comment. CivilComments is also used in Koh et al. [2021].

Table 1: Xie et al. [2021] propose In-N-Out (self-training) to mitigate ID-OOD accuracy tradeoffs—their method requires lots of unlabeled data. Even without this unlabeled data, ID-calibrated ensembles are competitive with or outperform self-training ID and OOD. We show results on all datasets used by Xie et al. [2021].

| | Cropland | | Landcover | | CelebA | |
| --- | --- | --- | --- | --- | --- | --- |
| | ID Acc | OOD Acc | ID Acc | OOD Acc | ID Acc | OOD Acc |
| Standard model | 95.3 (0.0) | **85.6 (5.8)** | 76.9 (0.3) | 55.7 (1.1) | 90.4 (0.5) | 74.5 (0.6) |
| Robust model | 95.1 (0.1) | 89.8 (0.4) | 72.7 (0.2) | **60.4 (1.1)** | **94.5 (0.2)** | 76.3 (1.2) |
| Self-training | 95.3 (0.2) | **90.6 (0.6)** | **77.0 (0.4)** | **61.0 (0.7)** | 93.1 (0.2) | **78.7 (0.7)** |
| Cal ensembling | **95.6 (0.1)** | **91.3 (0.8)** | **77.2 (0.2)** | 60.8 (0.8) | **94.5 (0.5)** | 77.6 (1.2) |

## 6 RESULTS

In Section 6.1, we show that ID-calibrated ensembles get the best of both worlds across the 11 natural shifts we consider, but not on the 3 adversarially synthesized anticorrelated spurious shifts, as predicted by our analysis in Section 4. ID-calibrated ensembles match or outperform a prior state-of-the art approach based on self-training [Xie et al., 2021], which requires additional unlabeled data. In Section 6.2, we show ablations of our method. Interestingly, we find that a common approach of tuning the ensemble weights to optimize ID accuracy can lead to poor OOD performance.

### 6.1 MAIN RESULTS

**Competitive with self-training.** Xie et al. [2021] propose self-training on unlabeled data to mitigate ID-OOD accuracy tradeoffs. We run experiments on all 3 datasets they consider (Landcover, Cropland, CelebA), taking checkpoints from the official CodaLab implementation of Xie et al. [2021]. Table 1 shows that ID-calibrated ensembles match or outperform self-training on all 3 of their datasets, both ID and OOD. We believe this is interesting because our method is simple and does not need additional unlabeled data (which, for example, the other datasets do not have).

**Strong ID and OOD accuracy.** Across the 11 natural shifts, ID-calibrated ensembles get the best of both worlds, typically outperforming the standard and robust model both ID (Table 2) and OOD (Table 3). Averaged across the *natural shift* datasets, ID-calibrated ensembles get 90.3% ID (vs. 88.7% for the standard model and 86.8% for the robust model) and 74.5% OOD (vs. 72.3% for the robust model and 65.2% for the standard model). The method works across the board—ID-calibrated ensembles achieve the best performance on 8/9 ID natural shifts, and on 10/11 OOD natural shifts. For the remaining two cases, DomainNet OOD and CIFAR-10 ID, ID-calibrated ensembles close over 95% of the gap between the standard and robust model.

**Shift type is important.** Our analysis in Section 4 predicts that ID-calibrated ensembles *do not* work as well on anticor-

related spurious shifts, where a spurious feature is correlated with the label but anticorrelated OOD. Indeed, in these cases the OOD accuracy of ID-calibrated ensembles is between the standard and robust model (Table 3). Even so, averaged across all 14 datasets ID-calibrated ensembles do well and get 90.0% ID (vs. 88.6% for the standard model, 86.9% for the robust model) and 74.7% OOD (vs. 64.3% for the standard model, 74.6% for the robust model).

### 6.2 ABLATIONS

Our proposed method is a simple combination of a calibrated robust and calibrated standard model. We vary the components of our method and try: (i) tuned ensembles without calibration, (ii) vanilla ensembles without calibration, and (iii) ensembles of two standard or two robust models.

**Tuned ensembles do not mitigate tradeoffs.** A natural way to ensemble the two models is "tuned ensembles": choosing the ensemble weights to optimize accuracy on the ID validation set. This approach is also known as stacking, and has performed well on the Netflix prize and Kaggle competitions [Sill et al., 2009]. Interestingly, we find that tuned ensembles do not do very well OOD, getting an average accuracy of 72.1% across the 14 datasets (vs. 74.7% for ID-calibrated ensembles). The ID accuracies are similar—results for all datasets are in Table 4 (ID) and Table 5 (OOD).

**Calibration helps.** ID-calibrated ensembles (calibration is only done on ID data) outperform vanilla ensembles, getting an average ID accuracy of 90.0% (vs. 89.4% for vanilla ensembles) and an average OOD accuracy of 74.7% (vs. 73.1% for vanilla ensembles). We show results for all datasets in Table 4 (ID) and Table 5 (OOD).

**Outperforms standard and robust ensembles.** As a sanity check, Appendix B.2 shows that our method outperforms 1. ensembling two (calibrated) standard models, and 2. ensembling two (calibrated) robust models.

**Models are miscalibrated OOD.** Even after ID calibration, we find that the standard and robust models are not calibrated OOD, which matches prior work [Ovadia et al.,

Table 2: *In-distribution (ID) accuracies for the standard model, robust model, and ID-calibrated ensembles, across 9 natural shift datasets (colored blue) and 3 anticorrelated spurious shift datasets (colored red and starred). On the 9 ID natural shift datasets, ID-calibrated ensembles match or outperform the best model in 8/9 cases, and on average outperforms both the standard and robust models. For the remaining dataset, CIFAR-10, ID-calibrated ensembles close 97% of the gap between the standard and robust model.*

|              | Ent30       | DomNet      | CIFAR10     | Liv17       | Land        | Crop        | CelebA      |
|--------------|-------------|-------------|-------------|-------------|-------------|-------------|-------------|
| Standard     | **93.6 (0.2)** | 83.9 (1.0) | **97.4 (0.1)** | 96.9 (0.1) | **76.9 (0.3)** | 95.3 (0.0) | 90.4 (0.5) |
| Robust       | 90.7 (0.2)  | 89.2 (0.1)  | 92.0 (0.0)  | 97.0 (0.0)  | 72.7 (0.2)  | 95.1 (0.1)  | **94.5 (0.2)** |
| Cal Ensemble | **93.7 (0.1)** | **91.2 (0.7)** | 97.2 (0.1) | **97.2 (0.2)** | 77.2 (0.2) | **95.6 (0.1)** | **94.5 (0.5)** |

|              | ImageNet    | iWildCam    | MNLI*       | Waterbirds* | CivilComments* |
|--------------|-------------|-------------|-------------|-------------|----------------|
| Standard     | 81.7 (-)    | 82.4 (-)    | **82.9 (-)** | 88.3 (-)   | **92.8 (-)**   |
| Robust       | 68.4 (-)    | 81.8 (-)    | 81.5 (-)    | **93.2 (-)** | 86.3 (-)      |
| Cal Ensemble | **82.0 (-)** | **84.0 (-)** | 82.8 (-)   | 92.9 (-)    | 91.4 (-)       |

Table 3: *Out-of-distribution (OOD) accuracies for the standard model, robust model, and ID-calibrated ensembles, across 11 natural shift datasets (colored blue) and 3 anticorrelated spurious shift datasets (colored red and starred). On the 11 OOD natural shift datasets, ID-calibrated ensembles match or outperform the best model in 10/11 cases, and on average outperforms both the standard and robust models. For the remaining dataset, DomainNet, ID-calibrated ensembles close 96% of the gap between the standard and robust model. As expected from our analysis (Section 4), on anticorrelated spurious shifts the accuracy of ID-calibrated ensembles is between the standard and robust models.*

|              | Ent30       | DomNet      | STL10       | Liv17       | Land        | Crop        | CelebA      |
|--------------|-------------|-------------|-------------|-------------|-------------|-------------|-------------|
| Standard     | 60.7 (0.1)  | 55.3 (0.4)  | 82.4 (0.3)  | 77.7 (0.6)  | 55.7 (1.1)  | **85.6 (5.8)** | 74.5 (0.6) |
| Robust       | 63.2 (1.1)  | **87.2 (0.1)** | 85.1 (0.2) | **82.2 (0.2)** | **60.4 (1.1)** | 89.8 (0.4) | 76.3 (1.2) |
| Cal Ensemble | **64.7 (0.5)** | 86.1 (0.2) | **87.3 (0.2)** | 82.2 (0.6) | **60.8 (0.8)** | **91.3 (0.8)** | **77.6 (1.2)** |

|              | ImNet-R     | ImNet-V2    | ImNet-Sk    | iWildCam    | MNLI*       | Waterbirds* | Comments*   |
|--------------|-------------|-------------|-------------|-------------|-------------|-------------|-------------|
| Standard     | 52.4 (-)    | 71.5 (-)    | 40.5 (-)    | 61.1 (-)    | 65.5 (-)    | 60.4 (-)    | 56.8 (-)    |
| Robust       | 77.5 (-)    | 61.9 (-)    | 48.2 (-)    | 63.0 (-)    | **77.4 (-)** | **88.1 (-)** | **84.2 (-)** |
| Cal Ensemble | **77.9 (-)** | **73.2 (-)** | **52.3 (-)** | **66.3 (-)** | 73.2 (-)   | 81.1 (-)    | 71.8 (-)    |

2019]. We estimate the expected calibration error (ECE; Equation 2 in Guo et al. [2017]). Since we calibrated on ID data, the ECE is low ID (1.6% for the standard model, 2.3% for the robust model; Table 8). However, the ECE is high OOD (11.3% for the standard model, 6.8% for the robust model; Table 9) Appendix B.3 shows that even the relative confidence of the models can be wrong: the standard model can be *more confident* but *less accurate* OOD, after ID-calibration. Nonetheless, ID-calibrated ensembles get the best of both worlds—see Section 4 for some simple intuitions for why this can happen.

## 7 RELATED WORKS AND DISCUSSION

**Calibration.** Calibration has been widely studied in machine learning [Naeini et al., 2014, Guo et al., 2017, Kumar et al., 2019], and applications such as meteorology [Murphy, 1973, DeGroot and Fienberg, 1983, Gneiting and Raftery, 2005], fairness [Hebert-Johnson et al., 2018], and healthcare [Jiang et al., 2012]. Many of these works focus on the in-distribution (ID) setting, where models are calibrated on the same distribution that they are evaluated on. Ovadia et al. [2019], Jones et al. [2021] show that if we calibrate (e.g., via temperature scaling) a model ID, it still has poor uncertainties OOD. However, we show that despite having poor uncertainties on traditional metrics, calibrated models can be combined effectively to mitigate ID-OOD tradeoffs. Wald et al. [2021] show that if a model is calibrated on many domains (domains > no. of features) in a linear setting, then the model is calibrated (and invariant) on new domains. A key difference is that they require a large number of training domains, which may need to be annotated to ensure calibration across them, while we only require access to a single doamin.

**Ensembling.** Ensembling models is a common way to get an accuracy boost—typically the ensemble members are trained with a different random seed [Lakshminarayanan et al., 2017] or augmentation [Stickland and Murray, 2020]. In the setting where the ensemble members mostly differ by random seeds or augmentations, prior work has shown that calibrating the members of an ensemble does not help [Wu and Gales, 2021, Ovadia et al., 2019]. However when we combine two very different models (standard and robust), calibration leads to clear improvements.

**Mitigating ID-OOD tradeoffs.** Tradeoffs between ID and OOD accuracy are widely studied and prior work self-trains on large amounts of unlabeled data to mitigate such tradeoffs [Raghunathan et al., 2020, Xie et al., 2021, Khani and Liang, 2021]. In contrast, our approach uses no extra unlabeled data and is a simple method where we just add up the model probabilities after a quick calibration step. In concurrent and independent work, [Wortsman et al., 2021] show that there *exists* a way to combine a CLIP zero-shot and fine-tuned model to get good ID and OOD accuracy—however learning how to combine the models may require OOD data, which is not available. We show that the natural way to learn how to weight ensemble members—selecting the weights to optimize ID accuracy—does not get the best of both worlds. In addition, their approach does not directly apply to settings where the standard and robust models have different architectures, such as In-N-Out [Xie et al., 2021].

**Conclusion and Future Work.** In this paper, we show that ID-calibrated ensembles, a simple method of calibrating a standard and robust model only on ID data and then ensembling them, can eliminate the tradeoff between in-distribution (ID) and out-of-distribution (OOD) accuracy on a wide range of natural shifts. We hope that this leads to more widespread use and deployment of robustness interventions.

ID-calibrated ensembles were competitive with prior work that used self-training, despite being simpler and not using additional unlabeled data. However, self-training may have advantages: we believe self-training may potentially eliminate tradeoffs even in anticorrelated spurious settings—it could be interesting for future work to compare ensembling and self-training theoretically, and see if their benefits are complementary. Additionally, ID-calibrated ensembles require twice the compute of a single model (although for fairness, we compared with an ensemble of standard or robust models), while self-training gives us a single model. One potential future direction is to see if ID-calibrated ensembles can be distilled into a single model (without additional unlabeled data).

## 8 ACKNOWLEDGEMENTS

This work was in part supported by the Open Philanthropy Project and NSF Award Grant No. 1805310, and NSF IIS 2045685. AR was supported by an Open Philanthropy Project AI Fellowship. We would like to thank Robbie Jones and the anonymous reviewers for helpful comments on our draft.

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
