# OpenReview forum: "Calibrated ensembles can mitigate accuracy tradeoffs under distribution shift"
_auai.org/UAI/2022/Conference — UAI 2022 Poster_

### Official Review · Reviewer_xikE · 2022-03-31

**Q2(1) Originality/Novelty:** 3
**Q2(2) Significance/Impact:** 3
**Q2(3) Correctness/Technical Quality:** 3
**Q2(6) Clarity Of Writing:** 4
**Q6 Overall Score:** 7
**Q8 Confidence In Your Score:** 3

**Q1 Summary And Contributions:**

We are often interested in obtaining models that yield good performance within a data domain but simultaneously generalize to out of domain data.
The paper proposes a simple yet effective approach: An ensemble of two calibrated models, one fitted for in-domain performance and a robust model that generalizes to new domains. The paper now claims that, if both models are calibrated and ensembled, the resulting model can improve in-domain and out-of-domain performance over individual models.



**Q2 Assessment Of The Paper:**

More detailed information regarding each of these aspects is given below:

**Q2(4) Quality Of Experiments (Optional):**

4: Excellent: The experimental evaluation is comprehensive and the results are compelling.

**Q2(5) Reproducibility:**

3: Good: Key resources (e.g., proofs, code, data) are available and key details (e.g., proofs, experimental setup) are sufficiently well-described for competent researchers to confidently reproduce the main results.

**Q3 Main Strengths:**

The proposed method seems simple and effective and the paper communicates it clearly.

The paper clearly defines the proposed problem, the proposed solution and provides intuition for why the observed improvements could occur. It furthermore benchmarks the proposed approach on a wide variety of datasets and settings.

The paper is well written and provides good intuition as to how observed effects might come to be. The experimental evaluation is well organized. The research question and operationalization in the benchmark are clear and well thought out.

The benchmark is sufficiently large and covers a variety of datasets and scenarios.

The detail of steps given in the different proofs is to be complimented. They are mostly easy to follow due to detailed steps and provided comments.

**Q4 Main Weakness:**

It is not clear to me why only parts of the WILDS benchmark are used instead of all datasets. The compared baselines only include the standard model and the robust model. It did not become clear from the text to me whether the baselines are the calibrated models or the un-calibrated models. This might provide an additional baseline and point of reference.

The paper does not provide any code that would enable reproducibility or a deeper investigation of the results.
This is problematic in so far as results seem almost a little too good.
Furthermore, some details on the experimental setup are missing (number of replications, hyperparameters, ...) which might be needed for replication.

The theoretical work shows proofs for a special case of the setting, i.e. balanced classes and assumption of orthogonality (4.1) this should perhaps be discussed more thoroughly.



**Q5 Detailed Comments To The Authors:**

Proofs of Prop 4.1 assume that r and s are orthogonal, which coincides with Assumption 4.1. I am not sure this assumption generally holds, so additional investigation regarding this might provide interesting insights.

Minor Points

I think Eq. (2.1) is unnecessary and does not generally hold in practice, as errors depend on the data (assume e.g. small n in one domain) and magnitude of shifts.


Typos:
Sec 4.1 calibrate fstd and frob ID
Sec 4.2 [on] OOD
Sec 6.2 citation of kaggle competitions is [?]
Sec 5: ? use fine tuning


**Q7 Justification For Your Score:**

I think the paper is solid and proposes a simple yet effective approach towarrds solving an important problem.
The experimental evaluation is interesting and covers serval important aspects.

**Q9 Complying With Reviewing Instructions:**

1: Yes.

---

### Official Review · Reviewer_GTAG · 2022-04-13

**Q2(1) Originality/Novelty:** 3
**Q2(2) Significance/Impact:** 3
**Q2(3) Correctness/Technical Quality:** 4
**Q2(6) Clarity Of Writing:** 3
**Q6 Overall Score:** 8
**Q8 Confidence In Your Score:** 4

**Q1 Summary And Contributions:**

This a simple and intuitive paper that proposes a way to improve both ID and OOD accuracy on many standard tasks. The idea is uniformly ensemble a robust and a standard model and this is shown to be help under certain kinds of shifts from train to test. The method is hyperparameter-free and plug-and-play and showcases good results.



**Q2 Assessment Of The Paper:**

More detailed information regarding each of these aspects is given below:

**Q2(4) Quality Of Experiments (Optional):**

4: Excellent: The experimental evaluation is comprehensive and the results are compelling.

**Q2(5) Reproducibility:**

3: Good: Key resources (e.g., proofs, code, data) are available and key details (e.g., proofs, experimental setup) are sufficiently well-described for competent researchers to confidently reproduce the main results.

**Q3 Main Strengths:**

The experimental results are great and the theoretical support is clean and intuitive. A second lesson here to me here is that the success of the proposed method also says something about the kind of real world shifts that exist (their method does not improve OOD if real-world shifts are adversarial like in the anticorrelated setting, as the authors point out).

**Q4 Main Weakness:**

I'm not going to argue against their results, which are extensive and good. my only concerns are with the theoretical results which are nice to and intuitive but seem to rely on weird assumptions. I think a few clarifications are warranted.

**Q5 Detailed Comments To The Authors:**


1. In prop 4.1, it is very weird that f_ens is better ID. I don't think this is possible if f_std is trained on ID data to maximize performance. The reason here is that when f_std learns the ID p( Y | X ), then f_std is a sufficient statistics and P_ID( Y | f_std, X) = P_ID( Y | f_std) meaning that f_std should be strictly better than f_ens and f_rob. Can the authors clarify why this is not the case?

2. Out of the three shifts considered, I am most sold on  the anticorrelated one; this is also the case where the method should not be expected to work. This is also the case where you cannot mitigate trade-off (nor should you try to). This seems to limit the applicability of the method because I'm not sure how we would know where to apply the method. Would I be able to choose whether to apply the method or now without access to labelled test data?

3. In the missing spurious features assumption, I'm not sure whether f_std would be zero. Wouldn't latching onto the shaggy mane mean f_std would say predict the other class? The suppressed spurious features assumption also seems to have the same problem, where less prevalence does not mean the conditional has lesser predictive probability (for the max probability class). Why would f_rob get affected by less prevalent country-specific features?

**Q7 Justification For Your Score:**

Experimental results are great and the method is very simple, both very good in my book. Only concern is justification of the assumptions, which is important in OOD.

AFTER REBUTTAL: some of the assumptions were clarified

**Q9 Complying With Reviewing Instructions:**

1: Yes.

---

### Official Review · Reviewer_mBc5 · 2022-04-13

**Q2(1) Originality/Novelty:** 2
**Q2(2) Significance/Impact:** 3
**Q2(3) Correctness/Technical Quality:** 3
**Q2(6) Clarity Of Writing:** 3
**Q6 Overall Score:** 6
**Q8 Confidence In Your Score:** 3

**Q1 Summary And Contributions:**

The paper explores a simple method in the distributional shifting setting, which is to ensemble the standard and robust models. It finds that after calibrating the model on only ID(in-distribution) data, it could outperform the baseline methods on both ID and OOD(out-of-distribution) under certain assumptions of the shifting. Theoretical results are given. Experiment results and ablation studies are conducted on several benchmark datasets.

**Q2 Assessment Of The Paper:**

More detailed information regarding each of these aspects is given below:

**Q2(4) Quality Of Experiments (Optional):**

2: Fair: The experimental evaluation is weak: important baselines are missing, or the results do not adequately support the main claims.

**Q2(5) Reproducibility:**

3: Good: Key resources (e.g., proofs, code, data) are available and key details (e.g., proofs, experimental setup) are sufficiently well-described for competent researchers to confidently reproduce the main results.

**Q3 Main Strengths:**

Intuitively it makes sense to me that since robust models and standard models could rely on different sets of features, ensemble them could make a better model. The method is easy to conduct and it performs well. Also, the paper is clearly written, and lots of theoretical and experimental results are shown. Ablation studies are conducted.

**Q4 Main Weakness:**

-While I like the idea of this paper, my main concerns lie in how practical are the assumption made in this paper, and how would the conclusion change if those assumptions do not hold? Such as what if the class balance assumption doesn’t hold? While assumption 4.1 is weaker than prior works, what if it doesn’t hold? Would these affect the conclusion made in the paper?

-Reading the intuition of why using calibration from the paper, it seems to me that this is because a simple ensemble method is used(½(f_std+f_rob)), but would another more well-fitted ensemble method make the calibration step not necessary?


**Q5 Detailed Comments To The Authors:**

-Related work: There is previous work discussing the relation between calibration and out-of-domain generalization. Although it differs from this paper in that it is in the multi-domain setting while this paper has one domain in training.
“On Calibration and Out-of-domain Generalization”, 2022

-minor typo: on Page 5&6, several citations related to lightweight fine-tuning seems not working

-Would be good to have the missing variance results on some of the datasets, especially on one of the anti-correlated one completed in Table 2&3.


**Q7 Justification For Your Score:**

While there are several issues that still remain under-explored, the paper proposes a simple method that works on several benchmarks in distribution shifting settings. And it provides new insights that could potentially have high impacts in the field.

**Q9 Complying With Reviewing Instructions:**

1: Yes.

---

### Official Review · Reviewer_F8r6 · 2022-04-15

**Q2(1) Originality/Novelty:** 2
**Q2(2) Significance/Impact:** 2
**Q2(3) Correctness/Technical Quality:** 3
**Q2(6) Clarity Of Writing:** 2
**Q6 Overall Score:** 5
**Q8 Confidence In Your Score:** 3

**Q1 Summary And Contributions:**

The authors propose to ensemble an ID model and an OOD model to achieve better performances in both settings. Under several assumptions, they provide several settings and theoretically analyze their method under these settings. Experiments on 11 datasets validate their theories.

**Q2 Assessment Of The Paper:**

More detailed information regarding each of these aspects is given below:

**Q2(4) Quality Of Experiments (Optional):**

2: Fair: The experimental evaluation is weak: important baselines are missing, or the results do not adequately support the main claims.

**Q2(5) Reproducibility:**

2: Fair: Key resources (e.g., proofs, code, data) are unavailable but key details (e.g., proof sketches, experimental setup) are sufficiently well-described for an expert to confidently reproduce the main results.

**Q3 Main Strengths:**

1. The paper provides an interesting solution to achieve better performance in both ID and OOD settings.
2. The paper is well-written and easy to follow.

**Q4 Main Weakness:**

1. The motivations of the assumptions / definitions are not clearly clarified.
2. The theories provide limited insight for designing a better OOD approach.
3. More baselines should be added.

**Q5 Detailed Comments To The Authors:**

I have several concerns about both theories and experiments.

1. The motivations of the assumptions / definitions are not clearly clarified.
   - For Assumption 4.1, I do not understand the relationship between the conditional independence constraint and the claim "$f_{\text{std}}$ relies on the spurious features while $f_{\text{rob}}$ relies on the robust features". In addition, I think $f_{\text{std}}$ should rely on both the spurious features and robust features. Moreover, a more detailed analysis of the related works should be provided.
   - For Definition 4.1, 4.2, and 4.3, the relationships between the mathematics forms and the motivation are also vague. Detailed analysis with concrete examples would help.
   - The authors develop the theories under the class-balanced assumption, which is strong in practice. Although the authors claim that they provide the general setting in Appendix A, I do not find any texts about this.
2. The theories provide limited insight for designing a better OOD approach. The paper assumes the availability of a robust model $f_{\text{rob}}$ and aims to ensemble it with the ID approach. In practice, we often need to train a robust model from the training data only. Moreover, we can not verify which kind of distribution shift could take place. As a result, we can not guarantee the effectiveness of the proposed method in real-world scenarios.
3. There are many OOD approaches and the authors should compare the results with them. To name a few, [1, 2].

Some minor issues

1. $T$ appears in the wrong place in both Equation 3.1 and Equation 3.2
2. There are many empty references (marked as ?) in the paper.

[1] Liu, Evan Z., et al. "Just train twice: Improving group robustness without training group information." International Conference on Machine Learning. PMLR, 2021.

[2] Nam, Junhyun, et al. "Learning from failure: De-biasing classifier from biased classifier." Advances in Neural Information Processing Systems 33 (2020): 20673-20684.

**Q7 Justification For Your Score:**

Due to limitations in both theories and experiments, I have a negative viewpoint of the paper in this round.

**Q9 Complying With Reviewing Instructions:**

1: Yes.

---

### Decision · Program_Chairs · 2022-05-15

**Decision:**

Accept (Poster)

**Comment:**

Meta Review: The reviewers found the paper to be of high-quality. The idea is simple to implement yet novel and insightful, the experimental evaluation is solid and the results are strong. The paper is well-written and gives good intuition about the results and their interpretation. During the review and discussion phases several questions and clarifications were made, and some additional experiments were promised by the authors - I trust these will be incorporated into the final accepted version of the paper (some of it possibly in the supplemental materials).